# Belief Embedding Tree Search

**Christopher Solinas**[1,†], **Radovan Haluška**[2,†], **Pavel Yanushonak**[2],
**David Sychrovský**[2], **Michael Buro**[1], **Martin Schmid**[2,3],
**Nathan R. Sturtevant**[1,4], **Michael Bowling**[1,4]

```
{solinas,mburo,nathanst,mbowling}@ualberta.ca,
{haluskr,sychrovsky,schmidm}@kam.mff.cuni.cz,
pavel.yanushonak351@student.cuni.cz
```

[1]**Department of Computer Science, University of Alberta, Canada**
[2]**Department of Applied Mathematics, Charles University, Prague**
[3]**EquiLibre Technologies, Inc.**
[4]**Alberta Machine Intelligence Institute (Amii)**

[†] Equal contributions.

## Abstract

We present Belief Embedding Tree Search (BETS), a novel planning algorithm for Partially Observable Markov Decision Processes (POMDPs). Effective planning hinges on accurately approximating the agent's belief state, yet existing methods become prohibitively expensive as belief states grow. BETS addresses this by compressing beliefs into fixed-length *embeddings* that are updated with each new observation. Conditioning a generative model on these embeddings enables Monte-Carlo planning over the approximate belief states. An initial evaluation in the standard benchmark *PocMan*—restricted to a reduced grid size—shows promising results compared to similarly budgeted particle filtering baselines. This highlights the potential for BETS to scale online planning to larger POMDPs.

## 1   Introduction

Computing posterior distributions over hidden states is crucial for planning in partially observable environments. Given observations, some planning algorithms for Partially Observable Markov Decision Processes (POMDPs) compute these *beliefs* with full Bayesian updates by exhaustively traversing the state space (Ross et al., 2008). Sample-based algorithms, such as Partially Observable Monte-Carlo Planning (POMCP) (Silver & Veness, 2010), scale better by sampling from approximate beliefs to evaluate candidate action sequences. However, in some cases, maintaining accurate belief approximations is computationally costly and dominates the algorithm's decision-time planning budget, leading to low search depth and breadth and suboptimal action selection.

Classical solutions approximate belief states with Bayesian filters: either lightweight with closed-form updates and strong parametric assumptions (e.g. (Kalman, 1960)), or non-parametric but computationally expensive particle filters (Doucet et al., 2009). Parametric filters with analytic posterior updates impose strong assumptions that are violated in tasks with multimodal belief states (Kalman, 1960; Julier & Uhlmann, 2004). Particle filters approximate arbitrary target distributions but can require an exponential number of particles (in the size of the state space) to avoid impoverishment and poor filtering performance (Thrun, 2002). Poor belief state approximation is well-known to have negative downstream effects on planning (Poupart & Boutilier, 2013).

Recently, Anonymous (2025) proposed Neural Bayesian Filtering (NBF), which models the set of beliefs induced by a task as a latent space. Latent vectors correspond to belief distributions and are

computed directly from samples. A generative model enables posterior computation in the latent space by simulating the environment one step upon a new observation. Crucially for planning, NBF approximates rich sets of belief distributions at a modest computational cost compared to particle filtering (Anonymous, 2025).

In this paper, we introduce Belief Embedding Tree Search (BETS). BETS is a planning algorithm that operates in a latent approximation of the belief states induced by a POMDP. It integrates NBF into a novel Monte Carlo planning algorithm, offering potentially significant scalability improvements over state-of-the-art approaches like POMCP. We demonstrate its performance empirically in PocMan (Silver & Veness, 2010), a standard POMDP planning benchmark.

## 2 Background

Before discussing the algorithm, we introduce some notation and concepts related to belief states and planning in POMDPs.

A **POMDP** is a tuple $G = (X, A, Y, T, H, R, \gamma)$. Here, $X$ is a set of discrete states, $A$ is a finite set of actions, and $Y$ is a set of observations. At each timestep the agent chooses an action $a \in A$ and moves from some state $x$ to $x'$, $x, x' \in X$. The transition function $T : X \times A \to \Delta X$ outputs the probability of transitioning to any $x \in X$, given a state-action pair. The observation function $H : X \times A \times X \to \Delta Y$ outputs the probability of all observations upon some transition. $R : X \times A \times X \to \mathbb{R}$ is the reward function that indicates an agent's reward upon a transition, and $\gamma$ is the discount factor. We assume $\gamma = 1$.

A **belief state** is a distribution over the Markov states at time $t$, conditional on actions and observations: $p_t(x) \overset{\text{def}}{=} p(x|a^{(1)}y^{(1)}, \ldots, a^{(t)}y^{(t)})$. In POMDPs, the agent chooses actions from belief states rather than *ground truth* states $x \in X$. Thus, an agent's policy $\pi$ is a mapping from belief states to distributions over $A$. In this paper, we consider only finite-horizon POMDPs, which means the agent's goal is choose a policy that maximizes expected reward $\mathbb{E}_{G,\pi}[\sum_{t=1}^{T} R(x_t, a_t, x_{t+1})]$ over episodes of the form $(x_1, a_1, x_2, a_2, \ldots, x_T, a_T)$. Planning algorithms interact with $G$ by performing simulations to select action sequences that approximate the optimal policy $\pi^*$.

**Particle filters** (Doucet et al., 2009) are non-parametric methods for approximating belief states. They represent arbitrary target distributions as sets of weighted *particles*. Upon receiving an observation, posterior beliefs are computed by simulating transitions in the POMDP for each particle and updating its weight accordingly. More concretely, given a policy $\pi$, action $a$, and observation $y$:

For each particle $x_i$ and particle weight $w_i$, $i \in 1, \ldots, n$:

1. Simulate transition $x_i' \sim T(x_i, a)$
2. Update weight $w_i' \leftarrow w_i \cdot \pi(x)[a] \cdot H(x_i, a, x_i')[y]$

When the weights of many particles become too small, particle filters typically *resample* by duplicating particles with higher probability and discarding others. Impoverishment occurs when too many particle weights become small and new particles cannot be resampled from outside $x_{1:n}$. This results in a degenerate approximation of $p_t(x)$ and significant errors in the expected reward estimates used for planning (Poupart & Boutilier, 2013). Protecting against impoverishment can require an exponential number of particles in the size of the state space (Thrun, 2002).

## 3 Belief Embedding Tree Search

PO-UCT performs simulations that require ground truth samples from the root belief state. POMCP (Silver & Veness, 2010) maintains an approximation of the relevant belief state with a particle filter. Methods for mitigating particle impoverishment exist, but are computationally expensive or only resample particles from within the originally sampled set (Sokota et al., 2022).

Neural Bayesian Filtering (NBF) (Anonymous, 2025) [1] mitigates impoverishment by computing posteriors in the embedding space of a belief embedding model. Each posterior update consists of

---

[1] NBF is under review. For further details, see the supplementary material.

---

**Algorithm 1:** Belief Embedding Tree Search

---

1 **Function** *Search*$(\theta, n, k)$
2     $\tau \leftarrow \texttt{InitTree}()$         // Track action values, visit counts, $\theta$
3     **for** $i = 1$ **to** $k$ **do**
4        $\theta_{\text{leaf}}, a_{1:t}, y_{1:t}, \tau \leftarrow \texttt{SimAndExpand}(\theta, \tau, n)$         // Add new leaf
5        $q \leftarrow \texttt{Evaluate}(\theta_{\text{leaf}}, n)$         // e.g. random rollouts
6        $\tau \leftarrow \texttt{Backup}(\tau, q, a_{1:t}, y_{1:t})$
7     **end**
8     **return** $\texttt{BestActionAtRoot}(\tau)$

9 **Function** *UpdateBeliefs*$(\theta, n, a, y)$
10    $x_{1:n} \leftarrow \texttt{NBF.generate}(\theta, n)$         // Run NBF to update and
11    $x'_{1:n}, w_{1:n} \leftarrow \texttt{NBF.update}(x_{1:n}, a, y)$     // weigh samples given $a$ and $y$
12    $\theta' \leftarrow \texttt{NBF.embed}(x'_{1:n}, w_{1:n})$
13    **return** $\theta'$

---

sampling particles from a generative model conditioned by the current belief embedding $\theta$, simulating them forward, and then computing a new weighted embedding from the result.

Belief Embedding Tree Search (BETS) (Algorithm 1) performs search in the embedding space. The algorithm has two main functions: returning an action given an embedding of a belief state, and updating a belief embedding given an action and observation. The function Search plans in the embedding space via PO-UCT to estimate the best action at the input belief state $\theta$. It uses a fixed budget of $k$ iterations and $n$ particles. UpdateBeliefs is called by SimAndExpand to compute embeddings for leaf nodes. After search, the agent executes the recommended action, receives an observation, and calls UpdateBeliefs to maintain beliefs for the next time it needs to act.

## 4 Experiments

We validate BETS in a small variant of PocMan (Silver & Veness, 2010). PocMan is a partially observable adaptation of the popular arcade game Pac-Man. Our version follows a similar design as Silver & Veness (2010), except we use fewer ghosts and power pills (2 instead of 4), a smaller grid (7x7 instead of 17x19) with 4 internal "pillars" as walls, and a randomized, unknown player location. Player and ghost locations are selected randomly every episode.

**Belief model training.** We train the belief embedding model for NBF *offline*. Trajectories are generated via biased random walks in the game, while a particle filter with 256 particles approximates the beliefs. Random walks are biased towards moving away from ghosts (except under the effect of the power pill) and towards food pellets. Particle sets from the particle filter are treated as samples from the target belief state and form a single training example in our training set. During training, each particle set is split, with half used to compute the embedding and the other half used to compute the training objective. The embedding network used to compute $\theta$ consists of 3 hidden layers with 128 units each. Its output is size 32 and conditions a Normalizing Flow (Papamakarios et al., 2021), with 5 coupling layers (Dinh et al., 2016) of size 32, for particle generation and a variational dequantization (Hoogeboom et al., 2020) layer to handle discrete data. The model is trained by maximizing the negative log-likelihood of the particle sets.

**Results.** We compare BETS against the baseline of using a particle filter in conjunction with PO-UCT (Silver & Veness, 2010). We label these baselines as **PF-**$n$, where $n$ is the number of particles. Table 1 shows the average reward of BETS compared to these baselines after 500 episodes, with standard error computed over 20 random seeds. A separate belief embedding model was trained and evaluated against the baselines for each random seed. PO-UCT was given simulation budgets of 50, 100, and 250 search iterations at every decision point and a maximum rollout depth of 100.

Table 1: Average Reward in 7x7 PocMan with no internal walls.

| PO-UCT iters. | PF-32 | PF-64 | PF-128 | BETS-32 | BETS-64 |
|---:|---|---|---|---|---|
| 50 | -0.09 ± 4.81 | 22.83 ± 5.22 | 50.23 ± 5.60 | 48.50 ± 6.54 | **84.01 ± 6.94** |
| 100 | 1.60 ± 4.94 | 24.05 ± 5.33 | 52.02 ± 5.64 | 50.13 ± 6.63 | **82.82 ± 6.91** |
| 250 | 1.45 ± 4.91 | 27.52 ± 5.47 | 54.20 ± 5.76 | 51.30 ± 6.60 | **86.23 ± 7.05** |

Increasing the number of particles helps particle filtering variants of PO-UCT perform better in planning across all simulation budgets. This is likely due to the effect of particle impoverishment. BETS matches all particle filtering variants of PO-UCT with only 32 particles and significantly outperforms them with 64 particles across all three simulation budgets. This demonstrates its potential and motivates further experiments in POMDPs with high-dimensional belief states.

## 5   Discussion and Future Work

BETS is a work in progress. The following are some key directions for future work related to algorithmic improvements and model training that could have a significant impact on performance.

**Online Belief Model Training.**   BETS currently requires a pre-trained belief embedding model to track the belief state with NBF. Data for this model is generated via random walks while tracking the belief state with a particle filter. Computational budgets for offline training are often significantly higher than decision time budgets (e.g., Silver et al. (2016)), so tracking the belief state using a more computationally expensive filter is valid, though vulnerable to impoverishment given long trajectories. Another issue is that the training dataset is generated by following a (biased) random walk; there is no guarantee that the belief states encountered during training are representative of those that will be reached by following a PO-UCT policy. Training, or fine-tuning, the belief model *online* by generating the relevant data with BETS has the potential to solve both of these problems.

**Policies and Value Functions on the Embedding Space.**   UCT (Kocsis & Szepesvári, 2006) and PO-UCT estimate leaf values with random rollouts and select actions based on action value estimates and visit counts. While extensions like PUCT (Rosin, 2011) and replacing random rollouts with neural network-based leaf evaluations led to landmark results in settings with fully-observable states (Silver et al., 2016), applying the same principles to particle-based approximations of belief states is not straightforward because it requires learning value functions and policies over sets of particles. Belief embeddings can potentially enable value and policy learning over large belief states.

**Model Architecture.**   This iteration of BETS copies the embedding and generative model architecture used in Anonymous (2025). Incorporating recent advances in generative modeling, such as Flow Matching (Lipman et al., 2022; Tong et al., 2023), and experimenting with different embedding architectures could significantly impact belief model accuracy and planning performance.

## 6   Conclusion

This paper introduced **Belief Embedding Tree Search** (BETS), a Monte-Carlo planning algorithm that replaces particle filters with Neural Bayesian Filtering to maintain compact, expressive beliefs during planning. On the small variant of PocMan we examined, BETS significantly outperformed or matched Particle Filtering PO-UCT with the same or higher particle budgets. Although our results are limited to small domains, they are promising in light of the algorithm's potential scalability. Future work will explore algorithmic improvements to BETS, and scale it to full-sized PocMan and other high-dimensional POMDPs.

**Acknowledgments.** This research was funded by the Canada CIFAR AI Chairs Program, and we acknowledge the support of the Natural Sciences and Engineering Research Council of Canada (NSERC). This research was also supported by the grant no. 25-18031S of the Czech Science Foundation (GAČR), Charles Univ. project UNCE 24/SCI/008 and the Charles University Grant Agency (GAUK), project no. 326525. Computational resources were provided by the e-INFRA CZ project (ID:90254), supported by the Ministry of Education, Youth and Sports of the Czech Republic.

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

# Supplementary Materials

*The following content was not necessarily subject to peer review.*

## 7 Neural Bayesian Filtering

Neural Bayesian filtering is an algorithm for tracking belief embeddings over a sequence of observations. It updates the input belief embedding by generating and then simulating $n$ particles for a single step. These particles are weighed according to the probability of input observation $y$ given the chosen action $a$, policy $\pi$, and environment dynamics.

---

**Algorithm 2:** Neural Bayesian Filtering

---

**input** : $\theta$ — Belief Embedding, $y$ — Observation, $a$ — Action, $(\psi, \phi)$ — Model Parameters, $n$ — Number of Particles, $\pi$ — Policy

**output:** $\theta'$ — Updated Belief Embedding

1 $z_{1:n} \overset{\text{iid}}{\sim} \mathcal{N}(0_d, I_d)$      `// n samples of d-dimensional Gaussian noise`

2 $x_{1:n} \leftarrow f_\psi(z_{1:n}; \theta)$      `// Generate particles from belief embedding`

3 **for** $i \leftarrow 1 \ldots n$ **do**

4     $x_i' \sim T(x_i, a)$

5     $w_i' \leftarrow w_i \cdot \pi(x)[a] \cdot H(x_i, a, x_i')[y]$

6 **end**

7 **return** $\mathcal{E}_\phi(x_{1:n}', \texttt{normalize}(w_{1:n}))$      `// Output embedding`

---

In addition to the input embedding $\theta$, observation, and action, the algorithm requires a generative model $f_\psi$ conditioned on $\theta$, and a permutation-invariant embedding function $\mathcal{E}_\phi$ that maps a set of weighted particles to an embedding. `UpdateBeliefs` in Algorithm 1 uses the same procedures to generate particles and updates embeddings for BETS.