# OpenReview forum: "Belief Embedding Tree Search"
_rl-conference.cc/RLC/2025/Workshop/RLVG — RLVG Workshop - RLC 2025_

### Official Review · Reviewer_8jqC · 2025-06-15
**Extending Neural Bayesian Filtering (NBF) to Planning**

**Rating:** 3
**Confidence:** 4

**Summary:**

The paper extends the recently submitted NBF algorithm to a planning setting, where the improvement in posterior estimation claimed by NBF is assumed to directly translate into improvements in the planning regime as well, particularly in terms of sample efficiency. The experiments are conducted in a POMDP setting based on the PocMan game and show that NBF+UCT outperforms the baseline of naive PF+UCT.

**Strengths:**

1. Presents an interesting integration of deep learning and particle filtering to address key limitations of naive Particle Filters, such as particle impoverishment and planning over high-dimensional belief distributions.

2. Experimental setup and hyperparameters are clearly described, enhancing reproducibility.

3. The paper is well-written and easy to follow.

**Weaknesses:**

1. Some claims in the original NBF paper, mentioned in the related works section, need to be empirically validated. For example, the claim that Deep Kalman Filters may not generalize well while NBF might. I believe the weights learned by NBF’s embedding space could face similar issues, unless empirically proven otherwise.

2. There are similar Deep Particle Filter works in the literature (for e.g., https://roboticsproceedings.org/rss14/p01.pdf) that might be worth discussing in the related works section.

**Best Paper Nomination:**

No

**Claims:**

Preliminary experiments do support their claim of improved sample efficiency over the naive Particle Filter baseline. However, a notable limitation lies in how the data is collected to train the embedding space, which may affect generalization beyond the specific training setup. That said, the authors are transparent about this limitation, which is appreciated.

**Suggestions:**

1. Consider including comparisons with variational and exact belief update methods from the literature, and highlight scenarios where the proposed approach offers an advantage—such as in handling multimodal belief states.

2. Incorporating more complex and diverse environments in the experiments could strengthen the empirical evaluation.

---

### Official Review · Reviewer_McLA · 2025-06-16
**Study of a tree search method with latent space belief embeddings**

**Rating:** 2
**Confidence:** 4

**Summary:**

The paper looks into a combination of a pre-trained model for belief state generation and a tree search method for online policy evaluation. The BETS + PO-UCT model is evaluated on a smaller version of the PocMan environment, showcasing improvement compared to the Particle filter + PO-UCT.

**Strengths:**

- The paper is concise and well written
- The study uses an established benchmark for POMDPs
- The experiments include several versions of PF (baseline) and BETS (new model), and their performance over different iteration budgets.

**Weaknesses:**

I appreciate the authors' comments and acknowledgement that this is a work in progress; however, there are several limitations:

My main concern is the simplification of the environment to a large degree (17x19 -> 7x7 + no internal walls). This appears to be a significant simplification of the underlying state space, and is related to the author's quote from the abstract: "This highlights the potential for BETS to scale online planning to larger POMDPs.".

The algorithm was tested and compared iteration-wise; it is not clear to me if the time complexity is comparable between the two methods. Adding an experiment with a set time budget would show the algorithm's performance in both sample efficiency and time complexity.

This method requires pre-training to generate the belief state; this does impact the off-the-shelf applicability of the method compared to the pure search-based algorithms. The details of the pre-training and the number of samples used for the random walk are not outlined in the paper. Assuming that a decent amount of data is required for the training of the NBF model, it would be fair to provide a comparison with a deep learning agent (DQN, PPO, etc) that is trained on a similar amount of data and a hand-crafted agent that uses the same heuristics as the random walk mentioned in the study.

**Best Paper Nomination:**

No

**Claims:**

Authors have provided sufficient evidence that, in the selected version of the PocMan environment, Neural Bayesian Filtering improves upon Particle Filtering.

**Suggestions:**

See weaknesses

---

### Decision · Program_Chairs · 2025-06-19

**Decision:**

Accept

**Comment:**

This paper contributes a novel planning algorithm that operates in a latent approximation of the belief states of POMDPs. The reviewers highlighted the clarity of the manuscript, the overall soundness of the method, and the interesting results presented. However, the authors point out the lack of ablations regarding time complexity, lack of some baselines and some missing references. We strongly encourage the authors to address these points, as well as the reviewers’ suggestions, in the camera-ready version.